# *Drosophila* as a Model of Unconventional Translation in Spinocerebellar Ataxia Type 3

**DOI:** 10.3390/cells11071223

**Published:** 2022-04-04

**Authors:** Sean L. Johnson, Matthew V. Prifti, Alyson Sujkowski, Kozeta Libohova, Jessica R. Blount, Luke Hong, Wei-Ling Tsou, Sokol V. Todi

**Affiliations:** 1Department of Pharmacology, Wayne State University School of Medicine, Detroit, MI 48201, USA; gg5745@wayne.edu (S.L.J.); ga5392@wayne.edu (M.V.P.); fs1484@wayne.edu (A.S.); klibohov@med.wayne.edu (K.L.); jrblount@wayne.edu (J.R.B.); lukehong@wayne.edu (L.H.); wtsou@wayne.edu (W.-L.T.); 2Department of Neurology, Wayne State University School of Medicine, Detroit, MI 48201, USA

**Keywords:** ataxin-3, Gal4-UAS, glia, Machado–Joseph disease, neuron, neurodegeneration, polyglutamine, RAN translation, RNA

## Abstract

RNA toxicity contributes to diseases caused by anomalous nucleotide repeat expansions. Recent work demonstrated RNA-based toxicity from repeat-associated, non-AUG-initiated translation (RAN translation). RAN translation occurs around long nucleotide repeats that form hairpin loops, allowing for translation initiation in the absence of a start codon that results in potentially toxic, poly-amino acid repeat-containing proteins. Discovered in Spinocerebellar Ataxia Type (SCA) 8, RAN translation has been documented in several repeat-expansion diseases, including in the CAG repeat-dependent polyglutamine (polyQ) disorders. The ATXN3 gene, which causes SCA3, also known as Machado–Joseph Disease (MJD), contains a CAG repeat that is expanded in disease. ATXN3 mRNA possesses features linked to RAN translation. In this paper, we examined the potential contribution of RAN translation to SCA3/MJD in *Drosophila* by using isogenic lines that contain homomeric or interrupted CAG repeats. We did not observe unconventional translation in fly neurons or glia. However, our investigations indicate differential toxicity from ATXN3 protein-encoding mRNA that contains pure versus interrupted CAG repeats. Additional work suggests that this difference may be due in part to toxicity from homomeric CAG mRNA. We conclude that *Drosophila* is not suitable to model RAN translation for SCA3/MJD, but offers clues into the potential pathogenesis stemming from CAG repeat-containing mRNA in this disorder.

## 1. Introduction

Long, tandem-repeating nucleotide sequences and resulting amino acid tracts are associated with various human genetic disorders [1,2,3]. When expanded, tandem repeats can detrimentally influence protein folding, function, and aggregation, resulting in repeat-expansion disorders [1,2,3]. In addition to anomalies stemming from changes at the protein level, expanded nucleotide repeats are also implicated in altering gene structure and causing RNA-mediated toxic effects [1,3]. One mechanism of RNA-based toxicity stems from repeat-associated, non-AUG-initiated (RAN) translation [2,4,5,6].

RAN translation allows for translation initiation and elongation through a repeat strand in the absence of an AUG initiation codon and can occur in various reading frames of a specific transcript [4,6]. This process enables the production of multiple homopolymeric or multi-amino acid repeat-containing proteins [4,6]. RAN translation was discovered by pioneering work in Spinocerebellar Ataxia Type 8 (SCA8) after the mutation of the ATG initiation codon in the CAG-repeat containing *ATXN8* gene did not block the expression of the polyglutamine (polyQ) repeat in transfected cells [2]. Upon further investigation, homopolymeric polyQ, polyalanine (polyA), and polyserine (polyS) proteins were found to be expressed in all three reading frames in the absence of AUG or near-cognate codons [2].

Although RAN translation was initially described in the context of CAG-repeat expansions in SCA8, it has since been shown to occur with the expansions of CAG, CUG, GGGCC, and CGG repeats associated with several repeat-expansion diseases [2,4,5,6,7,8,9,10,11,12,13]. These repeats drive RAN translation from within 5′ untranslated regions (UTRs), introns, and protein-coding open reading frames [2,4,5,6,7,8,9,10,11,12,13]. The exact mechanism through which RAN translation occurs remains to be clarified; however, as evidence of RAN translation expanded among various diseases, several common themes emerged. First, RAN translation is repeat-length dependent, with the likelihood of RAN-derived protein accumulation increasing with longer repeat tracts [4,6]. Second, long repeat sequences form secondary structures, including hairpin loops, which appear to be a requirement for RAN translation to take place—non-hairpin-forming repeats (such as those with CAA in place of CAG repeats) do not produce RAN proteins [4,6]. Third, the secondary structure of repeat-expanded RNA downstream of the start codon may recruit initiation factors and ribosomal subunits to internal ribosomal entry sites (IRESs), regardless of the presence of initiation or near-cognate initiation codons [4,6].

There are various repeat-containing disease proteins where RAN translation has yet to be identified as a factor in pathogenesis. One such example is another CAG-repeat expansion disease, SCA3, also known as Machado–Joseph Disease (MJD). SCA3/MJD is the most prevalent dominant ataxia worldwide and, along with Huntington’s disease, is among the most common of the family of nine polyQ repeat-expansion diseases [14,15,16,17,18,19,20,21,22]. It is caused by the expansion of the CAG repeat of the ATXN3 gene that encodes an expanded polyQ tract in ataxin-3, a deubiquitinase implicated in protein quality control and DNA repair [17,18,19,23,24]. Normally, these repeats exist in a range of 12–42, but expand to ~60–87 repeats in patients [14,15]. This long CAG repeat, similar to that of diseases such as SCA8 and Huntington’s disease where RAN proteins have already been detected, makes a SCA3/MJD an intriguing candidate for another potential instance of RAN translation.

mRNA toxicity was reported as a potential contributor in polyQ diseases, and more specifically SCA3/MJD [25,26,27,28,29]. The contribution of mRNA-based toxicity in SCA3/MJD was tested by altering the CAG repeat sequence of ATXN3 to have an alternating CAGCAA repeat, which significantly reduced toxicity in *Drosophila melanogaster* [25]. The expression of an untranslated version of the expanded CAG repeat also resulted in neuronal degeneration [25]. Additional studies suggested that the expanded ATXN3 gene is prone to frameshift mutations [30,31,32,33,34] and that these frameshifts result in the production of harmful polyA-containing proteins that have been observed in SCA3/MJD patient lymphoblasts and pontine neurons [30]. These studies were reported before the discovery of RAN translation; their conclusions may have been unrecognized evidence of the presence and influence of RAN proteins. The long CAG repeat, reliance on homomeric CAG repeat secondary hairpin structure for toxicity, and the presence of polyA proteins in SCA3/MJD patient tissue point to the potential of RAN translation in SCA3/MJD.

In this study, we set out to search for evidence of the occurrence of RAN translation in SCA3/MJD in vivo. We took advantage of the genetic versatility of *Drosophila melanogaster* models of SCA3/MJD to express versions of human ataxin-3 transgenes with the native homomeric CAG or interrupted alternating CAGCAA repeats that are, respectively, more and less likely to form the secondary structure necessary to enable RAN translation. Although we were able to identify the expression of polyA proteins when these constructs were expressed in cultured mammalian cells, we found no evidence of RAN proteins in any of our fly models of SCA3/MJD. We also returned to the idea of mRNA-based toxicity by expressing ATXN3 constructs that cannot be translated and found that the presence of a homomeric CAG repeat was, albeit to a small degree, more toxic than the interrupted repeat. Overall, this study points to a version of SCA3/MJD pathogenesis that experiences little-to-no input from RAN translation and apparent but limited contributions from mRNA toxicity in *Drosophila*, and underscores limitations with the *Drosophila* model system in studying RAN translation in this specific disease.

## 2. Materials and Methods

### 2.1. Construct Design

The ATXN3-Q80(CAGCAA) cDNA was based on the human ATXN3 sequences used in previous publications [35,36,37,38,39,40,41,42,43,44,45,46]. ATXN3-Q80(CAG) cDNA and the Met-Null cDNA counterparts to both ATXN3-Q80 constructs were designed from the ATXN3-Q80(CAGCAA) sequence and synthesized using the company Genscript (genscript.com, accessed on January 2, 2019). The transgenes were subcloned into pWalium-10.moe and transgenic fly lines were generated with the phiC31 integrase system into attP2 on chromosome 3 at Duke University Model Systems [38,39,40,41,42,43,45,47,48,49,50]. All insertions were confirmed and validated with PCR, genomic sequencing and then Western blotting using procedures described in this paper and previous work [39,40,41,42,43,45,47,51].

### 2.2. Antibodies

Anti-ataxin-3: rabbit polyclonal, 1:10,000 [52]; mouse monoclonal 1H9, 1:1000, Millipore. Anti-HA: rabbit monoclonal, 1:1000, Cell Signaling Technology. Anti-MYC: mouse monoclonal 9E10, 1:1000, Santa Cruz Biotechnology; mouse monoclonal 9B11, 1:1000, Cell Signaling Technology. Anti-V5: rabbit monoclonal D3H8Q, 1:1000, Cell Signaling Technology.

### 2.3. Mammalian Cells and Assays

Ataxin-3 constructs containing a pure CAG repeat or an interrupted CAGCAA repeat were sub-cloned from pWalium-10.moe (above) into pcDNA 3.1(+1) for expression in a mammalian cellular environment. M-17 or HEK-293T cells were purchased from ATCC (Manassas, Virginia), confirmed for lack of mycoplasma contamination and cultured under conventional conditions at 37 °C and 5% CO_2_ in DMEM supplemented with 10% FBS and 5% penicillin-streptomycin (Thermo Fisher, Waltham, MA, USA). Cells were transfected as indicated in the Results section using Lipofectamine LTX (Thermo Fisher). Then, 24 h later, cells were harvested for protein extraction and Western blotting. Cells were scraped with boiling cell lysis buffer (50 mM Tris pH 6.8, 2% SDS, 10% glycerol, 100 mM dithiothreitol), sonicated, boiled for 10 min, centrifuged at 13,300× *g* at room temperature for 10 min and loaded onto SDS-PAGE gels (4–20% gradient gels, Bio-Rad, Hercules, CA, USA).

### 2.4. Drosophila Stocks and Husbandry

The control fly line, isogenic host strain attP2 (#36303), and the MYC positive control line, a MYC-tagged ataxin-3 (#33610), were obtained from the Bloomington *Drosophila* Stock Center (Bloomington, Indiana). Stocks obtained as gifts were as follows: sqh-Gal4 (Dr. Daniel P. Kiehart, Duke University), elav-Gal4-GS (Dr. R. J. Wessells, Wayne State University), and elav-Gal4 and repo-Gal4 (Dr. Daniel F. Eberl, University of Iowa). The fly line used as the V5-positive control was a V5-tagged ataxin-3 Q80 line that we used in a previous publication [40]. The expression pattern of elav-Gal4 (pan-neuronal), repo-Gal4 (all glial cells, except for the midline glial cells), and sqh-Gal4 (ubiquitous) have been well-documented and reported in various publications [53,54,55,56,57,58,59,60,61,62,63,64,65,66,67].

Stocks and crosses were maintained at 25 °C and approximately 60% humidity in diurnal incubators with 12 h light/dark cycles. Flies were kept on a conventional cornmeal diet when being kept as stocks as well as throughout crosses and experimentation. The one exception to the standard diet was the glutamine-substituted (L-Glutamine; Millipore Sigma, Burlington, MA, USA) food that is described in the Results section. All flies were heterozygous for both driver and transgene, unless otherwise noted.

### 2.5. Quantitative Real-Time Polymerase Chain Reaction

Five intact adults or ten flies per biological sample (N) collected during development, depending on the experiment, had total RNA extracted using TRIzol (Life Technologies, Carlsbad, CA, USA). The extracted RNA was then treated with TURBO DNAse (Ambion) and reverse transcription was carried out using a high-capacity cDNA reverse transcription kit (ABI). Finally, mRNA levels were quantified with the StepOne Real-Time PCR system using a Fast SYBR Green Master Mix (ABI). The primers used for the original ATXN3-Q80 constructs were:

ATXN3 F: 5′-GAATGGCAGAAGGAGGAGTTACTA-3′;

ATXN3 R: 5′-GACCCGTCAAGAGAGAATTCAAGT-3′;

rp49 F: 5′-AGATCGTGAAGAAGCGCACCAAG-3′;

rp49 R: 5′-CACCAGGAACTTCTTGAATCCGG-3′

Primers used for Met-Null constructs:

Met-Null + ATXN3 F: 5′-GGCGATGCTTAGAGCGAAGA-3′;

Met-Null + ATXN3 R: 5′-AATCGAGACCGAGGAGAGGG-3′;

rp49 F: 5′-AGATCGTGAAGAAGCGCACCAAG-3′;

rp49 R: 5′-CACCAGGAACTTCTTGAATCCGG-3′

### 2.6. Drosophila Examinations

For longevity experiments, adults were collected on the day of eclosion and monitored daily to record deaths. Flies were switched into fresh food every other day. Males and females were tracked separately, unless otherwise noted. The number of adults tracked in each experiment (N) is noted in figures and legends. For developmental death tracking, flies were observed from the embryo stage through eclosion or death, and deaths at each developmental stage were recorded daily.

### 2.7. Western Blotting and Quantification

Western blots were performed with either 3 whole adults or 5–10 flies collected during development, per biological sample (N), depending on the experiment and driver being used. Samples were homogenized in boiling fly lysis buffer (50 mM Tris pH 6.8, 2% SDS, 10% glycerol, 100 mM dithiothreitol (DTT)), briefly sonicated, boiled for 10 min and centrifuged for 8 min at 13,300× *g* at room temperature. PXi 4 (Syngene, Frederick, Maryland) or ChemiDoc (Bio-Rad) were used to develop the Western blots, which were then quantified with GeneSys (Syngene) or ImageLab (Bio-Rad), respectively. Quantification was conducted using the volume of whole lanes measuring ataxin-3 protein levels and corrected for its own background. Signal measured included the main ataxin-3 band and all other ataxin-3 species above it. The ataxin-3 signal from each lane was then divided by its own loading control (direct blue staining, whole lane signal measurement) and reported as arbitrary units. Direct blue stains of total protein were performed by submerging the PVDF membranes for 10 min in 0.008% Direct Blue 71 (Sigma-Aldrich, St. Louis, MO, USA) in 40% ethanol and 10% acetic acid and then rinsed with a solution of just 40% ethanol/10% acetic acid, before being air dried and imaged.

### 2.8. Filter-Trap Assays

Three intact adults or five flies per biological sample collected during development, depending on the driver being used, were homogenized in a 200 µL of NETN buffer (50 mM Tris, pH 7.5, 150 mM NaCl, 0.5% Nonidet P-40) that was supplemented with a protease inhibitor (PI; S-8820, Sigma-Aldrich). The resulting lysates were then diluted with 200 µL 0.5% SDS in PBS. Diluted lysates were sonicated, centrifuged at 4500× *g* for 1 min at room temperature and then diluted further by combining 100 µL lysate with 400 µL PBS. A total of 35 µL of this final lysate of each sample were added to a Bio-Dot apparatus (Bio-Rad) and filter-vacuumed through a 0.45 µm nitrocellulose membrane (Schleicher and Schuell, Keene, New Hampshire) that had been pre-incubated with 0.1% SDS in PBS. After samples were filter-vacuumed through the membrane, the membrane was rinsed twice with 0.1% SDS in PBS, incubated in primary and then secondary antibody, and analyzed via Western blotting.

### 2.9. Co-Immunopurification Assays

Ten intact adults or fifteen developing flies per biological sample, depending on the driver being used, were homogenized in 400 µL of RIPA (50 mM Tris, 150 mM NaCl, 0.1% SDS, 0.5% deoxycholic acid, 1% NP40, pH7.4) + PI buffer, sonicated and then mixed with an additional 400 µL of RIPA + PI. Samples were then tumbled at 4 °C for 30 min, then centrifuged at 4 °C for 10 min at 10,000× *g*. While tumbling, bead-bound antibodies (either MYC- or V5-tagged; Thermo Fisher Scientific, Waltham, MA, USA) were prepared with three rinses of RIPA + PI. Following centrifugation, 40 µL of supernatant was combined with 10 µL of 6×SDS, boiled for 10 min and then saved as an input. The remaining supernatant was combined with the bead-bound antibodies and tumbled at 4 °C for 2 h. The beads were then rinsed 3 times with RIPA + PI with a 5 min tumble at 4 °C during each rinse. Beads then underwent 5 additional rinses with RIPA + PI without additional tumbling. Finally, bead-bound complexes were eluted by combining with 30 µL of lysis buffer (50 mM Tris pH 6.8, 2% SDS, 10% glycerol, 100 mM DTT) and boiling for 5 min.

### 2.10. Statistics

Statistical tests used for each experiment are stated in their respective figure legends. Prism 8 (GraphPad, San Diego, CA, USA) was used for the Kaplan–Meier log-rank tests for survival analysis, the Welch’s *t*-tests for RQ comparisons from qRT-PCRs, the unpaired *t*-test for Western blot analysis, and RM two-way ANOVA with Geisser–Greenhouse correction and Tukey’s multiple comparisons test for developmental death stage analysis. Additional data collection, organization, and Student’s *t*-tests were performed in Excel (Microsoft Version 16.55, Redmond, Washington) or Numbers (Apple Version 11.2, Cupertino, CA, USA). *p*-Values were calculated by the software used for analysis and are shown in the corresponding figures and legends, along with the number of independent biological replicates.

## 3. Results

### 3.1. Development of New Constructs to Examine RAN in SCA3/MJD

We generated two plasmids for mammalian cell expression that contain the full, human ataxin-3 sequence with either a pure homomeric CAG repeat (ATXN3-Q80(CAG)) or an interrupted, alternating CAGCAA repeat (ATXN3-Q80(CAGCAA)). Both produce the same pathogenic 80Q tract in the translated ataxin-3 protein; however, the pure CAG repeat is susceptible to the formation of abnormal/slippery hairpin loops and could lead to RAN translation, while the CAGCAA mRNA is less likely to produce hairpin loops or RAN peptides and precludes the production of proteins in alternative reading frames [4,6,25,26,27,28,32,68,69]. Constructs were HA-tagged in the normal, sense frame (encoding ATXN3 protein). MYC and V5 tags were added in the +1 and +2 sense reading frames, respectively (Figure 1A). Our expectation was that these tags would allow us to biochemically detect the presence of any possible RAN translated polyserine (polyS) or polyalanine (polyA) proteins from the sense strand.

The ATXN3-Q80(CAGCAA) fly line has been employed in our previous studies and no specific protein signal was observed in either the MYC- or V5-tagged reading frames [39,40,41]. Upon the transfection of cultured, mammalian HEK-293T cells with either ATXN3-Q80(CAG) or ATXN3-Q80(CAGCAA), we observed no specific protein signal from alternate reading frames with ATXN3-Q80(CAGCAA) and its alternating repeat. However, the transfection of ATXN3-Q80(CAG) possessing homomeric CAG repeats resulted in the detection of polyA proteins translated from the +2 V5-tagged frame (Figure 1B). As SCA3/MJD is a primarily neuronal disease, transfections were also conducted in a human neuroblastoma (M-17) cell line producing similar results (Figure 1B). These results provide evidence that RAN translation is possible with the expanded ATXN3 transcript. Our next step was to determine whether this phenomenon occurs in vivo.

### 3.2. Homomeric and Alternating Pathogenic Ataxin-3 PolyQ Repeats Are Differentially Toxic in Drosophila melanogaster

To investigate SCA3/MJD RAN translation in vivo, we again used *Drosophila melanogaster* [39,40,41,42,43,45,70,71]. Utilizing the versatility of *Drosophila* genetics along with the Gal4-UAS binary system of expression, we can express either the homomeric or the alternating polyQ repeat of pathogenic ataxin-3 transgenes individually in specific tissues [39,40,41,42,43,45,47,51,70,72,73]. We focused our investigations on the type of tissue that is primarily impacted in SCA3/MJD, neuronal and glial [14,15,18,74]. We used Gal4 drivers that express constructs in all neurons (elav-Gal4) or in all glial cells (except midline glia; repo-Gal4), and employed the same constructs as in Figure 1, except that the Kozak sequences were optimized for fly-based translation.

The expression of each construct was validated via quantitative real-time polymerase chain reaction (qRT-PCR). In flies pan-neuronally expressing the transgenes individually, ATXN3-Q80(CAG) was present at lower levels than ATXN3-Q80(CAGCAA) (Figure 2A). However, no significant differences in mRNA levels were detected between the two when expressed in glial cells (Figure 2B). Western blot analyses of samples obtained by lysing adult flies expressing either ATXN3-Q80(CAG) or ATXN3-Q80(CAGCAA) pan-neuronally (Figure 2C) or in developing flies expressing either version only in glia (Figure 2D) showed no significant difference in ataxin-3 protein level between the two transgenes.

We next determined if the presence of homomeric CAG repeats, and thus the potential for hairpin loop formation and RAN translation, resulted in differences in disease progression compared to the alternating, CAGCAA repeats. We performed longevity experiments with flies expressing either construct pan-neuronally. Survival analyses from males and females revealed that ATXN3-Q80(CAG) led to significantly decreased longevity compared to those expressing the alternating ATXN3-Q80(CAGCAA) in either sex (Figure 2E). In males, particularly, flies expressing the alternating repeat lived as long as 46 days, while none of their homomeric repeat-expressing counterparts was able to emerge as adults, dying as pharate adults instead (Figure 2E; Appendix A).

The expression of ataxin-3 in all glia, regardless of the transgene used, was developmentally lethal and no adults emerged for longevity studies (Figure 2F). The experiments conducted in Figure 2B,D were conducted with flies collected during development from lines expressing either construct in glia. Although we were unable to study differences in adult longevity between our two lines in glia, the high levels of toxicity in these cells compared to neurons points to an interesting role for glia in SCA3/MJD pathogenesis and is an active field of investigation; a recent study identified impaired oligodendrocyte maturation as a contributor to pathogenesis in a transgenic mouse model of SCA3/MJD [74].

In addition to pan-neuronal and glial expression, we performed survival studies in flies expressing either of the transgenes ubiquitously. This is an expression model that we suspected would be developmentally lethal and we thus focused on tracking the stages of development when flies died, similar to what we did in previous work [41]. As predicted, no adult flies emerged—they all died during development. Developmental observations yielded a similar pattern of results as our other expression models: we noted that ATXN3-Q80(CAG)-expressing flies died earlier during development than ATXN3-Q80(CAGCAA)-expressing ones (Appendix A).

These data indicate that, although the sequence of each ATXN3 transgene encodes the same pathogenic 80Q ataxin-3 protein with comparable protein levels, the presence of homomeric CAG is more toxic than the alternating repeat. This opens the possibility that factors outside of the full ataxin-3 protein may influence SCA3/MJD pathogenesis. To determine if RAN translation is one of those factors, we next examined the presence of proteins from alternative reading frames that could appear due to RAN translation.

### 3.3. Lack of Evidence of RAN Peptides in Drosophila from Pathogenic ATXN3

To detect the presence of potential RAN proteins, we took advantage of the alternate frame tagging of our ATXN3 constructs outlined in Figure 1. These tags allow us to probe for the polyS (+1 MYC-tagged frame) and polyA (+2 V5-tagged frame) proteins that we anticipate would be translated if RAN translation were occurring—particularly the polyA proteins that we observed in cultured mammalian cells expressing ATXN3-Q80(CAG). We conducted these probes using several biochemical assays in both neuronal and glial cell expression.

Simple Western blot analysis of whole fly lysates from lines pan-neuronally expressing either ATXN3-Q80(CAGCAA) or ATXN3-Q80(CAG) did not reveal any detectable, translated protein in either the +1 MYC-tagged or +2 V5-tagged frames (Figure 3A). As the aggregation of pathogenic ataxin-3 precedes toxicity in our *Drosophila* models of SCA3/MJD [39,40,41,42,43], we next wondered whether RAN species were trapped in larger aggregates, along with full-length ataxin-3, and thus may not have migrated properly into a standard SDS-PAGE gel. Thus, we conducted filter-trap assays designed to capture higher-order aggregated species [39,40,41,75,76,77]. There were no detectable MYC- or V5-tagged proteins in these assays (Figure 3B).

Finally, we conducted co-immunopurification (co-IP) assays targeting MYC- and V5-tagged proteins, in case the levels of these proteins were too low for capture by simple lysis of flies or through filter-traps. MYC and V5 beads were used in separate co-IP experiments. We again did not observe specific signal in either our MYC (Figure 3C) or V5 (Figure 3D) co-IPs.

These experiments were repeated in glial cells. Due to the developmental lethality of the glial expression of either ataxin-3 transgene in *Drosophila*, flies were collected during development for biochemical analysis, at either pupal or pharate adult stages. Western blot analyses recapitulated findings from pan-neuronal expression for all assays conducted (Figure 4).

We did not detect any proteins translated in alternative, sense frames, suggesting that, in *Drosophila*, RAN translation is not a factor in SCA3/MJD pathogenesis. The question still remains: why do we observe significantly higher toxicity from pathogenic ataxin-3 transgenes with homomeric CAG repeats compared to those with alternating CAGCAA tracts? To attempt to answer these questions, we turned to the previously theorized concept of contributions to ataxin-3 toxicity from its RNA [25,26,27,28,29].

### 3.4. New Transgenes to Investigate mRNA-Based Toxicity in Drosophila Models of SCA3/MJD

mRNA-based toxicity can occur in genes with long repeat sequences that allow for the formation of RNA foci; these foci can recruit mRNA away from their native functions, recruit mRNA-binding proteins, and can become toxic [29]. To investigate mRNA toxicity in fly models of SCA3/MJD ATXN3, we examined the impact of ATXN3 mRNA with homomeric CAG repeats versus CAGCAA repeats, while preventing the translation of ataxin-3 protein. We utilized the pathogenic ataxin-3 transgenes from the previous assays (ATXN3-Q80(CAG) and (CAGCAA)) and mutated all ATG (start) codons into TGA (stop) codons. These methionine-less constructs are referred to as Met-Null(CAG) and Met-Null(CAGCAA) throughout the rest of this work. They differ from their homomeric and alternating repeat, protein expressing, counterparts only at the ‘start’ to ‘stop’ mutations (Figure 5A). Western blots from flies expressing these transgenes in all neurons or in glial cells did not yield any specific signal in the ataxin-3 frame (Figure 5B,C).

### 3.5. Minimal Survival Difference in Flies Expressing Alternating or Homomeric, Met-Null CAG/A Repeats

We examined if differences in longevity between alternating and homomeric ataxin-3 CAG repeats observed in Figure 2 resulted in part from toxicity at the mRNA level. We expressed Met-Null(CAG) or Met-Null(CAGCAA) pan-neuronally in flies and examined their survival (Figure 6A, Appendix A). The analyses of these survival data revealed sex-specific differences in longevity in flies expressing both homomeric and alternating Met-Null constructs, with females outliving males in both cases (Figure 6A; a similar sex-specific trend was observed in Figure 2E). In addition, females flies pan-neuronally expressing Met-Null(CAGCAA) lived significantly longer than those expressing Met-Null(CAG) (Figure 6A). However, this result was reversed in male flies expressing either of the Met-Null constructs: males pan-neuronally expressing Met-Null(CAG) lived slightly, but significantly, longer than ones expressing Met-Null(CAGCAA) (Figure 6A). qRT-PCR of these flies showed no difference in pan-neuronal expression between the homomeric and alternating CAG repeat constructs (Figure 6B). We repeated the same assays in glial cells. Male flies expressing homomeric Met-Null(CAG) ATXN3 lived slightly shorter lives than those expressing alternating repeats, but this difference was not significant in females (Figure 6C, Appendix A). qRT-PCR confirmed that there were no significant differences in expression between the two transgenes in glia (Figure 6D). We conclude that mRNA toxicity is unlikely to be significant in SCA3/MJD pathogenesis.

### 3.6. Alternative Mechanisms of Homomeric CAG Toxicity

Recently, a hypothesis emerged that the translation of pure expanded CAG repeats in polyQ disorders depletes available glutaminyl charging tRNA^Gln-CUG^, which pairs exclusively with the CAG codon [78]. This depletion would result in a transcriptome that is prone to mistranslation and could account for the pathological differences observed between pure and interrupted CAG transcripts [78]. This notion provided a potential mechanism to help to explain the differences we observed in our experiments between both the translated protein and Met-Null versions of ATXN3.

Our probe into this hypothesis focused on the availability of glutamine. In order for tRNA^Gln-CUG^ to be charged and contribute to the translation of long CAG repeats, there must also be an appropriate level of available glutamine to charge tRNA^Gln-CUG^. Perhaps glutamine is a bottleneck, decreasing in levels as CAG repeats become longer and causing toxicity as a result of depletion. We supplemented normal fly food with 0.00, 12.5, or 50.0 g of glutamine per liter of media to determine if increased dietary glutamine overcomes potential reductions in this amino acid. Concentrations for glutamine supplementation were based on previous studies of amino acid supplementation in flies [79,80].

We tested the effect of increasing amounts of available glutamine in the fly diet using the ubiquitous expression of ATXN3-Q80(CAG) and ATXN3-Q80(CAGCAA) that resulted in high toxicity (Appendix A). By tracking developmental deaths, we were able to assess if there was any benefit from increasing amounts of glutamine. Additional dietary glutamine did not provide any detectable benefit to developing flies expressing ATXN3-Q80(CAG) or ATXN3-80(CAGCAA) (Appendix A). These data suggest that glutamine itself is not a limiting factor in these models.

While glutamine supplementation does not appear to be the answer in overcoming the hypothesized tRNA depletion that results from expanded CAG repeat translation, there are additional components to the translation machinery that could still be tested. These components, including the tRNA charging machinery and the tRNA^Gln-CUG^ itself, are outside the scope of this work, but comprise viable candidates for future investigations.

## 4. Discussion

We aimed at identifying the presence and potential influence of RAN translation in fly models of SCA3/MJD. As with previously identified RAN translation-associated disorders, such as SCA8 and Huntington’s disease, SCA3/MJD is a CAG repeat-expansion disorder with a long repeat stretch. Based on prior reports on ATXN3 mRNA and that of other CAG repeat disorders [25,26,27,28,29], SCA3/MJD toxicity may be influenced by the ability of ATXN3 mRNA to form secondary structures and by the presence of polyA species that have been observed in SCA3/MJD tissue [30]. These studies pointed to the possibility of RAN translation and made SCA3/MJD a candidate for exploratory research to add it to the growing list of repeat-expansion diseases influenced by RAN translation.

We utilized transgenic *Drosophila* lines to express two versions of pathogenic ATXN3 in the tissues primarily affected in SCA3/MJD, neurons and glia. The two versions of ATXN3 had either an uninterrupted CAG or an alternating CAGCAA repeat in their polyQ-encoding tract and represented ATXN3 mRNA capable (pure CAG) or incapable (alternating CAGCAA) of forming secondary structures for RAN translation. Although we observed differences in toxicity between the two transgenes, we did not find biochemical evidence of RAN-translated proteins in *Drosophila*. This lack of evidence does not preclude the possibility of RAN translation occurring in the sense, ataxin-3 HA-tagged reading frame, or RAN products in the antisense strand. The detection of these possibilities is outside the capability of our current model systems; however, our work in mammalian cells and the presence of polyA proteins in SCA3/MJD patient tissue [30] support our focus on the aberrant translation of the polyS and polyA reading frames as a reasonable measurement of the occurrence of RAN translation in SCA3/MJD flies.

The absence of evidence of RAN translation in our SCA3/MJD model system led us to consider mRNA playing a possible role in ATXN3-based toxicity, as previously reported by the Bonini lab [25]. We tested this non-mutually exclusive alternative to RAN translation by using redesigned, pathogenic ATXN3 transgenes whose potential start codons were mutated to stop codons. The expression of these ‘Met-Null’ transgenes yielded minor differences in toxicity that were not consistent between sexes and between glia and neurons. These results provide some support for the possibility of mRNA-based toxicity in SCA3/MJD, as indicated previously [25], with the caveat that evidence in our study points to a highly subdued role for pure CAG mRNA toxicity in SCA3/MJD.

Differences in the extent of mRNA toxicity between our work and prior findings [25] could be explained by the potentially higher level of expression reached by prior transgenic lines (P element-based transgenesis that can lead to multiple insertions) compared to the single copy, safe-harbor integration of transgenes through the phiC31 integrase system that we employed. Additionally, earlier observations were reached by using isolated, hyperexpanded CAG expansions [25], instead of the patient-range used here. It is possible that mRNA toxicity does not feature prominently in the context of patient-range expansions, or that mRNA toxicity from isolated CAG repeats is stronger than when those repeats are in the context of ATXN3 sequence.

Various mechanisms may explain the contribution in toxicity by CAG repeat-containing mRNA. The uninterrupted CAG mRNA secondary structure could be more prone to aggregation than CAGCAA mRNA and may help to accelerate the aggregation of the ataxin-3 protein [6,68,69]. Homomeric mRNA could also be more prone to stalling at the ribosome, similar to other repeat-expansion diseases, triggering the dysregulation of essential processes [81]. In addition, these prolonged homomeric repeats could create shortages in the tRNA charging machinery that may further injure the cellular environment. Together with the toxicity stemming from the ataxin-3 protein itself, these mRNA-based mechanisms might hasten SCA3/MJD.

As stated above, while the models that we generated do not indicate the presence of RAN translation in SCA3/MJD flies, they do not exclude the existence of RAN-derived peptides in people. It remains to be established whether RAN translation does indeed occur in SCA3/MJD patients; once that is determined, patient-derived neuronal and glial cultures, alongside organoid-based examinations and mammalian models of disease, can detail the relative contributions of RAN-type translation in SCA3/MJD initiation and progression. Following this type of understanding, any future clinical interventions can be designed to include—or not—specific steps that address RAN-derived peptides in addition to ATXN3 protein-dependent toxicity. We conclude that *Drosophila* is not well suited for studying RAN-based toxicity for SCA3/MJD. Our study introduces to the field additional tools that can be utilized towards understanding SCA3/MJD and other similar disorders.

## Figures and Tables

**Figure 1 cells-11-01223-f001:**
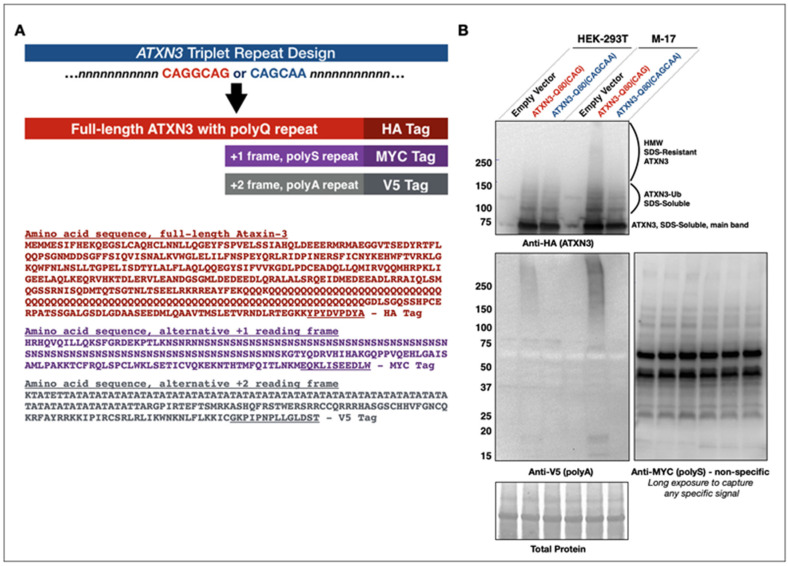
Designing ataxin-3 constructs for studying RAN translation. (**A**) Diagrammatic representation of the construct design with an uninterrupted CAG triplet repeat or alternating CAGCAA repeats and HA, MYC, and V5 tags in the ataxin-3, +1 and +2 reading frames, respectively. Additionally, the expected translated sequences of full-length proteins from each sense reading frame are shown. These proteins are expected to contain either polyglutamine, polyserine, or polyalanine repeats and their associated tags for identification. (**B**) Western blot from lysates of two different mammalian cell lines, HEK-293T or M-17, transfected with an empty pcDNA vector or a pcDNA-ATXN3-80(CAG) or pcDNA-ATXN3-80(CAGCAA). Transfection and lysis were performed independently for each cell line and then loaded together in one gel, as shown. Note the presence of bands between 15 and 20 kDa in the ‘CAG’ lanes, as well as the smears above 50 kDa in the same lanes that are absent in the other lanes.

**Figure 2 cells-11-01223-f002:**
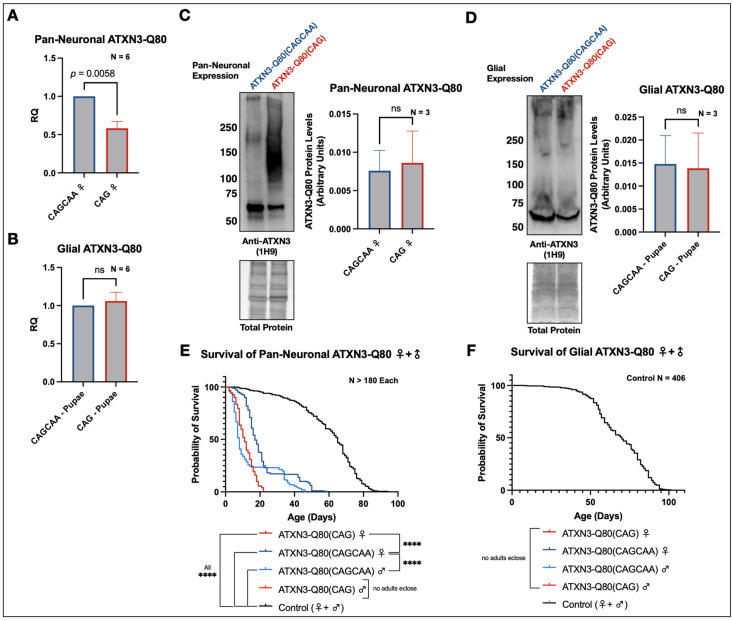
Homomeric CAG repeats are more toxic than alternating CAGCAA in fly models of SCA3/MJD. (**A**) qRT-PCR of one-day-old female adult flies pan-neuronally expressing ATXN3-80(CAGCAA) or ATXN3-Q80(CAG). Means ± SEM. *p*-value from Welch’s *t*-test. (**B**) qRT-PCR of developing flies expressing ATXN3-80(CAGCAA) or ATXN3-Q80(CAG) in glia. Means ± SEM. “ns”: non-statistically significant; Welch’s *t*-test. (**C**) Western blot of one-day-old adult female flies pan-neuronally expressing ATXN3-80(CAGCAA) or ATXN3-Q80(CAG) with whole-lane quantification of total ataxin-3 protein levels. Means ± SD. “ns”: non-statistically significant; unpaired *t*-test. (**D**) Western blot of developing flies expressing ATXN3-80(CAGCAA) or ATXN3-Q80(CAG) in glia with whole-lane quantification of total ataxin-3 protein levels. Means ± SD. “ns”: non-statistically significant; unpaired *t*-test. Developing flies were used because no adults emerge when either ATXN3 construct is expressed in glia. (**E**) Survival analysis from adult flies pan-neuronally expressing ATXN3-Q80(CAG), ATXN3-Q80(CAGCAA), or an empty vector insertion into the attP2 site (control). No adult male flies expressing ATXN3-Q80(CAG) eclosed as adults; they all died as pharate adults.; “****”: *p* < 0.0001. *p*-values from log-rank tests. (**F**) Survival analysis from adult flies expressing ATXN3-Q80(CAG), ATXN3-Q80(CAGCAA), or an empty vector insertion into the attP2 site (control) in glia. No adult flies ecloded in lines expressing either ataxin-3 transgene in glia; they all died as pharate adults. In panels A-D, ‘N’s represent independent biological samples/replicates.

**Figure 3 cells-11-01223-f003:**
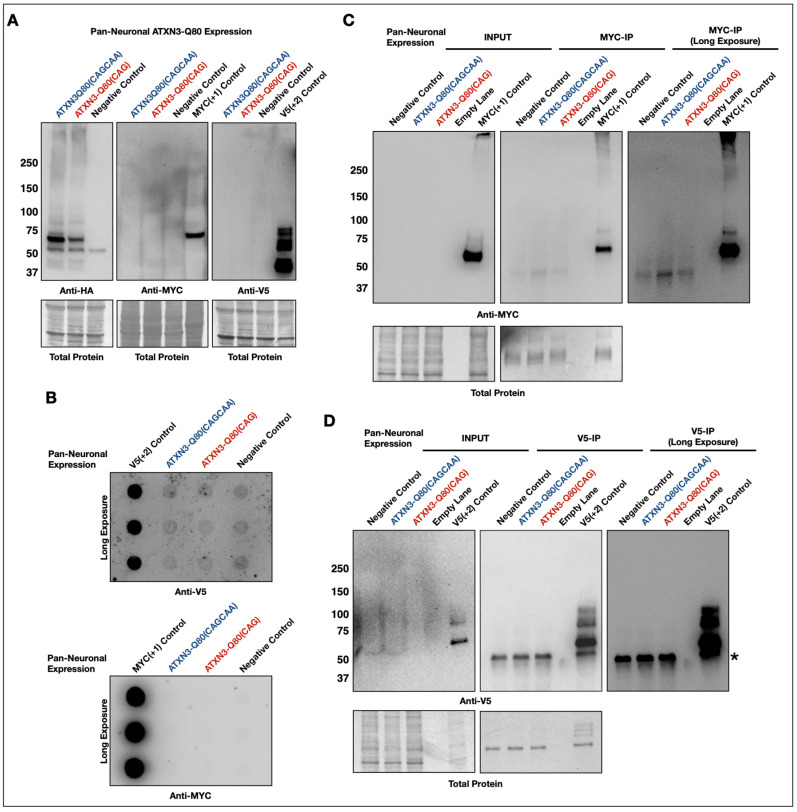
No evidence of RAN translation in pan-neuronal *Drosophila* models of SCA3/MJD. (**A**) Western blots of one-day-old adult female flies pan-neuronally expressing ATXN3-80(CAGCAA), ATXN3-Q80(CAG), or a negative control lacking any ataxin-3 transgene to examine MYC- or V5-tagged translated proteins from alternate reading frames. No specific signal was detected in the +1 or +2 reading frames. N ≥ 3 each. (**B**) Filter-trap assays of lysates from one-day-old adult female flies pan-neuronally expressing ATXN3-80(CAGCAA), ATXN3-Q80(CAG), or a negative control lacking any ataxin-3 transgene and blotting for either V5- or MYC-tagged proteins. V5- and MYC-blotted filter traps were conducted independently. No signal above background was detected in either case. N ≥ 3 each. (**C**,**D**) Co-immunopurification with anti-MYC (**C**) or anti-V5 (**D**) antibody-tagged beads of lysates from one-day-old female flies pan-neuronally expressing ATXN3-Q80(CAG) or ATXN3-Q80(CAGCAA). No alternate ataxin-3 frame MYC- or V5-tagged proteins were detected. N ≥ 3. Asterisk (*) denotes a non-specific band observed with the V5 antibody under some conditions. For all applicable panels, MYC-positive controls were from whole flies ubiquitously expressing a MYC-tagged ataxin-3 acquired from the Bloomington *Drosophila* Stock Center; V5-positive controls were from dissected heads of flies expressing a V5-tagged ataxin-3 in fly eyes that was used in a previous publication [40].

**Figure 4 cells-11-01223-f004:**
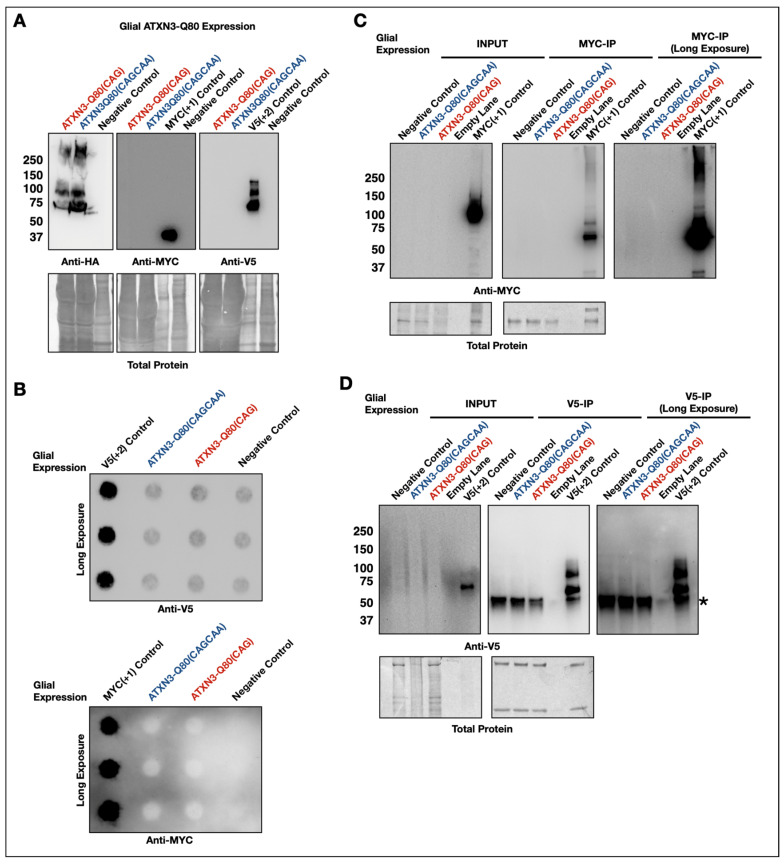
No evidence of RAN translation in the glia of fly models of SCA3/MJD. (**A**) Western blots of developing flies expressing ATXN3-80(CAGCAA), ATXN3-Q80(CAG), or a negative control lacking ATXN3 in glia. No specific signal was detected in the +1 or +2 reading frames. N ≥ 3 each. Developing flies were used for glial expression as no adults emerge when either ATXN3 is driven by repo-Gal4. (**B**) Filter-trap assays of lysates from developing flies expressing ATXN3-80(CAGCAA), ATXN3-Q80(CAG), or a negative control in glia. No signal above background was detected in either reading frame. N ≥ 3 each. (**C**,**D**) Co-immunopurification with anti-MYC (**C**) or anti-V5 (**D**) antibody-tagged beads from lysates of developing flies expressing ATXN3-Q80(CAG) or ATXN3-Q80(CAGCAA) in glia. No alternate frame MYC- or V5-tagged proteins were co-immunopurified. N ≥ 3. Asterisk (*) denotes a non-specific band observed by the V5 antibody under some conditions. For all applicable panels, MYC-positive controls were from whole flies ubiquitously expressing MYC-ataxin-3; V5-positive controls were from the heads of flies expressing a V5-ataxin-3 in fly eyes.

**Figure 5 cells-11-01223-f005:**
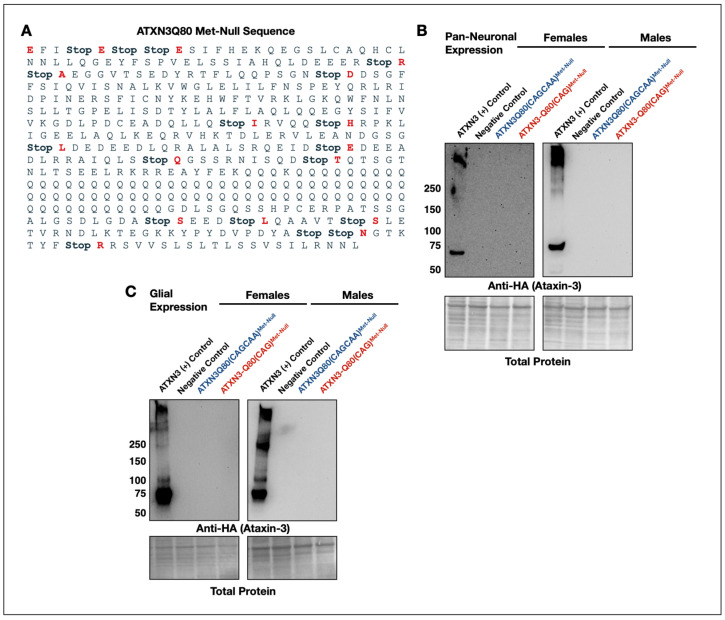
ATXN3 constructs to investigate mRNA-toxicity. (**A**) Each start codon was mutated into a stop codon (Stop) in ATXN3. Mutations were identical for the Met-Null(CAG) and Met-Null(CAGCAA) constructs, changing every ‘ATG’ into “TGA’. (**B**,**C**) Western blots of one-day-old adult female and male flies expressing Met-Null(CAGCAA), Met-Null(CAG), or a negative control lacking ATXN3 in all neurons (**B**) or glial cells (**C**). N ≥ 3 each. No ataxin-3 protein was detected. N ≥ 3 each. Nucleotide sequences for (**A**) are in Appendix A.

**Figure 6 cells-11-01223-f006:**
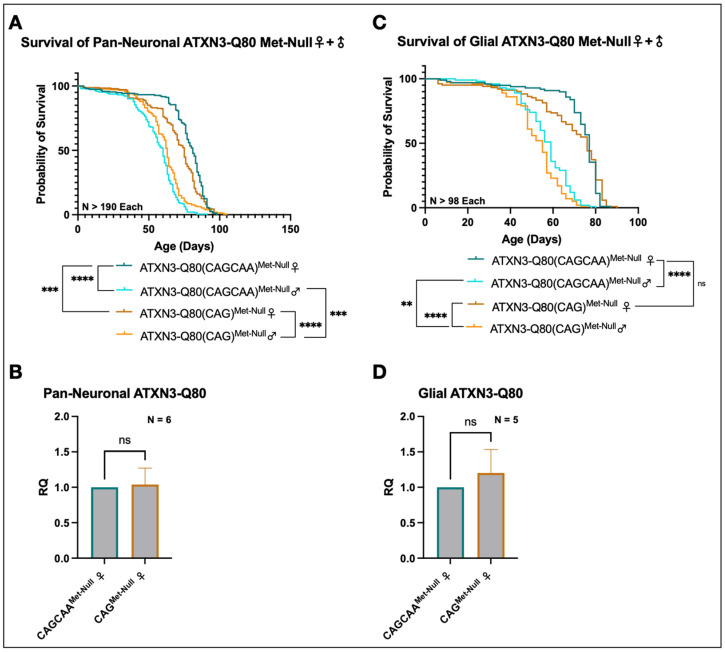
Homomeric CAG mRNA can be slightly more toxic than alternating CAGCAA in flies expressing Met-Null ATXN3. (**A**) Survival analyses from adult flies pan-neuronally expressing Met-Null(CAG) or Met-Null(CAGCAA). “***”: *p* < 0.0002; “****”: *p* < 0.0001. *p*-values from log-rank tests. (**B**) qRT-PCR of one-day-old adult female flies pan-neuronally expressing Met-Null(CAGCAA) or Met-Null(CAG). Means ± SEM. “ns”: non-statistically significant; Welch’s *t*-test. (**C**) Survival analyses from adult flies expressing Met-Null(CAG) or Met-Null(CAGCAA) in glial cells. “ns”: non-statistically significant; “**”: *p* < 0.0021; “****”: *p* < 0.0001. *p*-Values from log-rank tests. (**D**) qRT-PCR of one-day-old female flies expressing Met-Null(CAGCAA) or Met-Null(CAG) in glia. Means ± SEM. “ns”: non-statistically significant; Welch’s *t*-test. In (**B**,**D**), ‘N’s represent independent biological samples/replicates. Overall longevity of Met-Null flies for neuronal and glial expression approaches that of controls (e.g., Figure 2).

## Data Availability

No large-scale datasets were generated or analyzed for this study. All results pertaining to the figures and the text are included in the figures and Appendix A.

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
