# Peer review of "Drosophila as a Model of Unconventional Translation in Spinocerebellar Ataxia Type 3"

_cells, 2022, doi:10.3390/cells11071223_

Round 1

Reviewer 1 Report

I would like to start by congratulating the authors on this very interesting project, on the possible contribution of RAN translation to MJD/SCA3 in Drosophila 17 melanogaster. Even though the results of this project were negative, it is very important for the scientific community to have access to this report.

The paper is overall well written, the methods described in detail and the results well presented.

I would just have a few comments

  • What would be the authors’ plan on a future model to study the contribution of RAN translation in MJD/SCA3
  • What is the authors’ opinion on the importance of RAN translation in future treatment strategies
  • SCA3 is also recognized as Machado Joseph Disease (MJD). This designation should be maintained, and the manuscript corrected in accordance

Author Response

We appreciate the Reviewers’ assessment of our work and their input. We hope that our responses to each comment and the revisions in text (with tracked changes) fully address these points and render our paper acceptable for publication in Cells. 

We were extended 14 days to revise and resubmit; therefore, we assumed that no new experimental evidence was required to properly address each issue raised.

_____________________________________________________________________________________

I would like to start by congratulating the authors on this very interesting project, on the possible contribution of RAN translation to MJD/SCA3 in Drosophila 17 melanogaster. Even though the results of this project were negative, it is very important for the scientific community to have access to this report.

The paper is overall well written, the methods described in detail and the results well presented.

Response: Thank you for your comments and appreciation of our work.

I would just have a few comments

  • What would be the authors’ plan on a future model to study the contribution of RAN translation in MJD/SCA3

Response: Thank you for this insightful question. We have modified the last paragraph of the Discussion as follows to address this point: “As stated above, while the models that we generated do not indicate the presence of RAN translation in SCA3/MJD flies, they do not exclude the existence of RAN-derived peptides in people. It remains to be established whether RAN translation does indeed occur in SCA3/MJD patients; once that is determined, patient-derived neuronal and glial cultures, alongside organoid-based examinations and mammalian models of disease, can detail the relative contributions of RAN-type translation in SCA3/MJD initiation and progression. Following this type of understanding, any future clinical interventions can be designed to include — or not — specific steps that address RAN-derived peptides in addition to ATXN3 protein-dependent toxicity.

  • What is the authors’ opinion on the importance of RAN translation in future treatment strategies

Response: This is another key point, which we address in the prior response, underlined.

  • SCA3 is also recognized as Machado Joseph Disease (MJD). This designation should be maintained, and the manuscript corrected in accordance

Response: Thank you for this suggestion.We have modified the text to reflect this annotation. 

Reviewer 2 Report

In this paper, Johnson and colleagues investigate the role of RAN unconventional translation (repeat-associated, non-AUG-initiated translation) in the pathophysiology of Spinocerebellar Ataxia Type 3 (SCA3), a CAG-repeat expansion disease. The rationale behind the choice of addressing this topic is well discussed in the introduction and points to a real need for clarification of the role of this recently discovered type of RNA-based toxicity in this disease. To address this issue, they use as an animal model Drosophila Melanogaster, taking advantage of its genetic versatility, by expressing two different versions of the human ataxin3 transgene (with the human disease-mimicking Q80 repetition) which are supposed to be more or less prone to lead to RAN translation. However, RAN translation was not detected in this animal model despite clear effects on fly survival resulted from transgenes expression. To clarify these effects, the authors investigate other alternative mechanisms of CAG toxicity, but are unable to find a clear explanation.

Despite this research work would be in principle highly relevant to clarify the pathogenetic mechanisms behind SCA3, several points are not convincing and should be better addressed to make this paper suitable for publication.

  1. The use of Drosophila to investigate RAN translation is supported by the evidence that this animal model has already been used in the past to study this type of RNA-based toxicity in other disease models, such as FXTAS (Todd et al., 2013). Nevertheless, from the negative results obtained, the authors conclude that this animal model is not suitable to study RAN translation in SCA3, or, in general, to study RAN translation (lines 509-510). This sounds like an overstated conclusion. Rather, the constructs used in this study may not be suitable to produce relevant amounts of proteins in alternative reading frames and, thus, may not be suitable to mimic RAN translation. Indeed, the authors apply new constructs, that were first used in mammalian cell lines for validation of the method. However, results in these cell lines are overall weak. Western blots of the transfected cells are not convincing and should be supported by further evidence. The presumably positive band corresponding to polyA proteins translated from the +2 V5-tagged frame is very faint. Since the whole study is based on the validity of such constructs, the authors should perform immunostainings on the transfected cell lines to corroborate this result as done in other studies (Bañez-Coronel et al., 2016) and/or perform other experiments like those shown in figure 3.

For the in vivo experiments, several aspects appear not clear and should be better explained/addressed by the authors:

  1. The rationale behind the choice of expressing the constructs either in neurons or in glial cells is lacking. This should be somehow anticipated either in the introduction or in the corresponding result section.
  2. Along the same line, the method used by the authors for this tissue-specific expression is not explained. This should be properly addressed in the Methods section. Confirmation of tissue-specific expression is also lacking and should be provided. Moreover, what do the authors mean for glia? This term includes several cell types (astrocytes, oligodendrocytes etc.) and this should be better clarified by the authors.
  3. From line 258 onwards, it is not clear why western blot analyses were performed on adult flies when the constructs are expressed pan-neuronally and on developing flies when dealing with glial expression. The reader only later understands that this is a matter of survival but this should be better clarified before in the text.
  4. Survival experiments show differences between males and females for both constructs. Nevertheless, it is not clear whether this is somehow an exacerbation of any existing difference already present in control flies or not. To clarify this point, the authors should show survival lines separately for control males and females.
  5. There is overall inconsistency in when the analyses were performed based on the animal model. When the constructs were expressed ubiquitously, survival experiments were performed during development, tracking the stages of development when flies died. This could have been done also for glial expression, but the authors claim they were unable to study differences in longevity between the two lines in glia. The same holds true for males with ATXN3-Q80(CAG) expression, that die before they become adult and are not shown in the survival curve. Since these flies are those showing the most dramatic phenotype, why did the author choose to perform the following experiments on adult females? This may have masked potentially interesting results.
  6. In figure 6A, the survival line for control flies is missing. Based on the one shown in figure 2E, the values shown here are likely to be not so different from those of controls. The authors should clarify this point and comment accordingly.

Other minor comments:

  • How were western blots quantified? What kind of normalization was performed? This should be better addressed in the Methods section.
  • There is inconsistency between the number of biological replicates indicated in Figure 2D (n=3) and what reported in the methods section (n=5-10)

Author Response

We appreciate the Reviewers’ assessment of our work and their input. We hope that our responses to each comment and the revisions in text (with tracked changes) fully address these points and render our paper acceptable for publication in Cells. 

We were extended 14 days to revise and resubmit; therefore, we assumed that no new experimental evidence was required to properly address each issue raised.

The attached PDF contains our responses with the included figure mentioned in our response to point #6.

_____________________________________________________________________________________

In this paper, Johnson and colleagues investigate the role of RAN unconventional translation (repeat-associated, non-AUG-initiated translation) in the pathophysiology of Spinocerebellar Ataxia Type 3 (SCA3), a CAG-repeat expansion disease. The rationale behind the choice of addressing this topic is well discussed in the introduction and points to a real need for clarification of the role of this recently discovered type of RNA-based toxicity in this disease. To address this issue, they use as an animal model Drosophila Melanogaster, taking advantage of its genetic versatility, by expressing two different versions of the human ataxin3 transgene (with the human disease-mimicking Q80 repetition) which are supposed to be more or less prone to lead to RAN translation. However, RAN translation was not detected in this animal model despite clear effects on fly survival resulted from transgenes expression. To clarify these effects, the authors investigate other alternative mechanisms of CAG toxicity, but are unable to find a clear explanation.

Despite this research work would be in principle highly relevant to clarify the pathogenetic mechanisms behind SCA3, several points are not convincing and should be better addressed to make this paper suitable for publication.

1. The use of Drosophila to investigate RAN translation is supported by the evidence that this animal model has already been used in the past to study this type of RNA-based toxicity in other disease models, such as FXTAS (Todd et al., 2013). Nevertheless, from the negative results obtained, the authors conclude that this animal model is not suitable to study RAN translation in SCA3, or, in general, to study RAN translation (lines 509-510). This sounds like an overstated conclusion. Rather, the constructs used in this study may not be suitable to produce relevant amounts of proteins in alternative reading frames and, thus, may not be suitable to mimic RAN translation. Indeed, the authors apply new constructs, that were first used in mammalian cell lines for validation of the method. However, results in these cell lines are overall weak. Western blots of the transfected cells are not convincing and should be supported by further evidence. The presumably positive band corresponding to polyA proteins translated from the +2 V5-tagged frame is very faint. Since the whole study is based on the validity of such constructs, the authors should perform immunostainings on the transfected cell lines to corroborate this result as done in other studies (Bañez-Coronel et al., 2016) and/or perform other experiments like those shown in figure 3.

Response: Thank you for raising these points and our ability to clarify them.

  • Regarding lines 509-510 (currently, lines 844-845), the text has been edited to focus specifically on SCA3/MJD. 
  • Regarding the V5 blot in figure 1, we have revised the figure legend to highlight that it is not just the band we are referring to, but also a smear of species at the higher potions of the gel (lines 336-338). In these cells, we observe clear evidence of these species that we never observed with fly-based work. Also, we want to reiterate that the constructs used in flies and in mammalian cells are exactly the same, with the exception of the Kozak sequence to allow for optimized fly translation. We further clarify this in the text, lines 347-349.

For the in vivo experiments, several aspects appear not clear and should be better explained/addressed by the authors:

2. The rationale behind the choice of expressing the constructs either in neurons or in glial cells is lacking. This should be somehow anticipated either in the introduction or in the corresponding result section.

Response: An excellent point that we address in lines: 345-349.

3. Along the same line, the method used by the authors for this tissue-specific expression is not explained. This should be properly addressed in the Methods section. Confirmation of tissue-specific expression is also lacking and should be provided. 

Response: Thank you for raising the need for additional clarification for this point. We further expand on this in the Methods section, lines: 194-197.

4. Moreover, what do the authors mean for glia? This term includes several cell types (astrocytes, oligodendrocytes etc.) and this should be better clarified by the authors.

Response: We have clarified this in lines added pertaining to points 2 and 3, to note that by glia we mean all fly glial cells—except for the midline glia. We hope this addresses this point. We have added additional references for this information. 

5. From line 258 onwards, it is not clear why western blot analyses were performed on adult flies when the constructs are expressed pan-neuronally and on developing flies when dealing with glial expression. The reader only later understands that this is a matter of survival but this should be better clarified before in the text.

Response: We regret the lack of clarity here and have modified the legends of figures 2 and 4 to note this information.

6. Survival experiments show differences between males and females for both constructs. Nevertheless, it is not clear whether this is somehow an exacerbation of any existing difference already present in control flies or not. To clarify this point, the authors should show survival lines separately for control males and females.

Response: We used combined sexes for controls since the differences between controls and Atxn3-expressing flies in figure 2 were so dramatic. There is a difference between the longevity of male and female flies (figure included in the attached PDF version of our responses), as has also been reported before by others and us. But, since there is no clear indication that there are sex differences in SCA3/MJD patients, we did not pursue this any further in this manuscript. 

7. There is overall inconsistency in when the analyses were performed based on the animal model. When the constructs were expressed ubiquitously, survival experiments were performed during development, tracking the stages of development when flies died. This could have been done also for glial expression, but the authors claim they were unable to study differences in longevity between the two lines in glia. The same holds true for males with ATXN3-Q80(CAG) expression, that die before they become adult and are not shown in the survival curve. Since these flies are those showing the most dramatic phenotype, why did the author choose to perform the following experiments on adult females? This may have masked potentially interesting results.

Response: We appreciate this comment and the opportunity to further clarify our work and its resulting data. We performed longevity studies with adult flies wherever possible. We have modified the text to address the following:

  • ubiquitous driver: we further highlight in the manuscript that expression of these transgenes through the ubiquitous driver, which drives expression throughout development and in adults, causes developmental lethality, thus we are unable to perform adult longevity. (Lines 315-317)
  • glial driver: all flies died as pharate adults, thus we were unable to perform longevity assays. We now clarify this in the text and legends (lines: 436-437)
  • neuronal driver: the males died as pharate adults. We now clarify this in the legend. Line 439.

8. In figure 6A, the survival line for control flies is missing. Based on the one shown in figure 2E, the values shown here are likely to be not so different from those of controls. The authors should clarify this point and comment accordingly.

Response: Our focus was on the difference between the two types of Met-Null constructs, rather than on their direct comparison to controls. If the Reviewer deems these data necessary, we will need to conduct additional studies, which will take longer than the 14 days that we were provided to submit revisions. However, based on the data we have on hand, these flies live similarly to controls we tested for other assays. We have modified the text to address this point (lines: 578-579).

Other minor comments:

  • How were western blots quantified? What kind of normalization was performed? This should be better addressed in the Methods section.

Response: We have included additional details in the Methods section, as follows: “Western blot quantification was conducted using the volume of whole lanes measuring ataxin-3 protein levels, corrected for its own background. Signal measured included the main ataxin-3 band and all other ataxin-3 species above it. Ataxin-3 signal from each lane was then divided by its own loading control (direct blue staining, whole lane signal measurement) and reported as arbitrary units.” Lines: 238-243.

  • There is inconsistency between the number of biological replicates indicated in Figure 2D (n=3) and what reported in the methods section (n=5-10)

Response: We apologize for any confusion. We have clarified the methods and legends to state that: 5-10 means individuals per sample for lysis purposes in the Methods section, whereas “N” in figures means actual biological replicates, i.e. 3 repeats with 10 heads each. Lines: 209, 232, 248, 267, 288.

Round 2

Reviewer 2 Report

The authors have addressed all the points raised by this reviewer and the text was modified accordingly. Although further experiments were asked in the first report, it is reasonable that 15 days were not sufficient to perform them. The clarifications provided by the authors concerning the criticisms raised were however satisfactory and now the paper is suitable for publication.